# Quantifying the Language Barrier—A Total Survey of Parents’ Spoken Languages and Local Language Skills as Perceived by Different Professions in Pediatric Palliative Care

**DOI:** 10.3390/children7090118

**Published:** 2020-09-01

**Authors:** Larissa Alice Dreier, Boris Zernikow, Julia Wager

**Affiliations:** 1Paediatric Palliative Care Centre, Children’s and Adolescents’ Hospital, 45711 Datteln, Germany; b.zernikow@kinderklinik-datteln.de (B.Z.); j.wager@deutsches-kinderschmerzzentrum.de (J.W.); 2Department of Children’s Pain Therapy and Paediatric Palliative Care, Faculty of Health, School of Medicine, Witten/Herdecke University, 58448 Witten, Germany

**Keywords:** communication, language barriers, palliative care, pediatrics, nursing

## Abstract

To date, there are no specific figures on the language-related characteristics of families receiving pediatric palliative care. This study aims to gain insights into the languages spoken by parents, their local language skills and the consistency of professional assessments on these aspects. Using an adapted version of the “Common European Framework of Reference for Languages”, the languages and local language skills of parents whose children were admitted to an inpatient pediatric palliative care facility (N = 114) were assessed by (a) medical staff and (b) psychosocial staff. Nearly half of the families did not speak the local language as their mother tongue. The most frequently spoken language was Turkish. Overall, the medical staff attributed better language skills to parents than the psychosocial staff did. According to them, only 27.0% of mothers and 38.5% of fathers spoke the local language at a high level while 37.8% of mothers and 34.6% of fathers had no or rudimentary language skills. The results provide important information on which languages pediatric palliative care practitioners must be prepared for. They sensitize to the fact that even within an institution there can be discrepancies between the language assessments of different professions.

## 1. Introduction

Communication is an integral component of health care worldwide. Looking at data on root causes for adverse events and patient harm in clinical practice, faulty communication is consistently among the top four ranks [1]. Of all communication channels, language has the greatest potential to be an obstacle to communication [2]. Language and culture form an inseparable, complex and mutually influential unit. Language is formed by the unique characteristics of a culture and at the same time gives expression to it. Comparable to a mirror, language therefore provides essential insights into the culture of a person [3,4]. Because of today’s cultural diversity, successful communication on the language level cannot be taken for granted either in general health care or in specialized disciplines like pediatric palliative care. In pediatric palliative care, successful communication is essential due to the following reasons: First, numerous care providers are involved in a child’s care, requiring a constant exchange between the various care providers and between different care providers and parents [5]. Second, patients with various diagnoses, distressing symptoms and disease progressions are treated [5,6]. Good communication between parents and multi-professional caregivers is essential to initiate the best possible treatment options for the severely ill child. Third, the family forms the basic unit of care. To actively involve the family in the identification of care priorities and needs they need full information regarding their child’s illness and treatment [7]. Finally, many families care for their child at home and are therefore primarily responsible for its care, which requires that medical and nursing issues are understood in order to apply them correctly [8,9,10].

The obstacles of language barriers in pediatric palliative care have been reported by caregivers, especially in the context of accurate and complete dissemination of information [11,12,13]. From the perspective of foreign families, language barriers may result in dissatisfaction with care [14]; difficulties in utilizing health care services [14,15] and reduced quality of care [16,17,18]. In the clinical routine in general and particularly in pediatric palliative care, different strategies such as the involvement of (professional) translators are used to overcome language barriers between care providers and patients/families [13,18,19,20,21,22,23,24,25]. The importance of the awareness and education of cultural characteristics is acknowledged by many healthcare providers, but is perceived as insufficient [26,27,28]. However, successful intercultural communication can only be achieved if providers develop culturally sensitive skills and hence act on the same linguistic and superordinate cultural level as parents and patients with a different ethnic background [22,29,30,31].

To the best of our knowledge, there are no concrete numbers of language-related characteristics of parents whose children receive palliative care. However, a thorough knowledge of the languages and language skills of parents can provide a first significant insight into the cultures encountered in pediatric palliative care and thus contribute to culturally sensitive care. Therefore, this study aims to answer the following questions: (1) Which languages do families admitted to a pediatric palliative care unit speak? (2) If the local language is not spoken in the mother tongue: How good are the mother’s and father’s local language skills? Furthermore, and finally, (3) do staff members of different professions agree in their assessment of parental local language skills?

## 2. Materials and Methods

### 2.1. Clinical Setting and Participants

The study was conducted on a pediatric palliative care unit which is affiliated to a pediatric hospital in a national conurbation of Germany. A multi-professional team annually cares for about N = 170 children and adolescents between 0–21 years with life-limiting diseases in a total of 8 patient rooms [32]. Additional services such as the occasional use of interpreters, especially for conversations between parents and professionals, complete the care spectrum.

The following study is a full survey of all parents of children who were admitted to the pediatric palliative care unit in the year 2018. No further inclusion or exclusion criteria needed to be defined. Ethical approval was obtained by the Ethics Committee of the Children’s and Adolescents’ Hospital Datteln (Approval Code: 2019/01/23/BZ). It was confirmed that obtaining informed consent was not necessary as parents were only assessed by the staff during their regular stay and did not have to provide any further information about themselves or their child. Collected data was pseudonymized.

### 2.2. Recording Sheet

A recording sheet to assess the spoken language(s) of each parent was compiled. If the parent did not speak the local language (German) as his or her mother tongue, the recording sheet furthermore assessed the parent’s local language skills. A version of the German “Common European Framework of Reference for Languages” (“Gemeinsamer Europäischer Referenzrahmen für Sprachen”) adapted by the research team was used for this purpose. The Common European Framework of Reference for Languages and its German version is divided into three superordinate competence levels which in turn can be subdivided into six language skills levels. In addition, the framework can be supplemented with intermediate levels if necessary. Each language skills level is explained by a short text describing what a specific person can do in terms of their language skills [33]. Figure 1 shows the adapted version of the framework as utilized in this study. In the adapted version, the framework was supplemented by two language skill levels which reflect the mastery of no or only minimal local language skills (A0, A0+), as these were not covered by the original version of the framework, and as clinical experience suggested that a certain percentage of parents would speak at these two language levels.

As an additional source of information, the assessment forms, which are routinely filled in by the parents when admitted to the pediatric palliative care unit, were reviewed, as they also ask for data on spoken languages.

### 2.3. Data Collection

During the study period, both parents were individually assessed by the two professions once during or shortly after their child’s inpatient stay on the unit. The assessment was based on the clinical contacts with the respective parent. Obviously, parents could not be assessed if they were not present during their child’s inpatient stay. To determine to what extent staff members of different professions agree in their judgement, an assessment of (1) a member of the medical staff (physicians, nurses) and (2) a member of the psychosocial staff (psychologist, pedagogue) was obtained for each parent. All staff members were native speakers of German (the local language).

### 2.4. Data Analysis

Data were descriptively analyzed. The local language skills assigned to mothers and fathers by the two professions and their respective frequencies (overall distribution) were determined. Each existing assessment made by the two professions was included in the analyses.

The concordance between the two professions was examined at the individual person level by means of a Kappa statistic to test interrater reliability. The ranges of kappa can be interpreted as follows: <0.00 (poor), 0.0020130.20 (slight), 0.21–0.40 (fair), 0.41–0.60 (moderate), 0.61–0.80 (substantial), 0.81–1.00 (almost perfect) [34]. Each parent for whom an assessment was available from both professions was included in the analyses. All statistical evaluations were made with the statistics software IBM SPSS Statistics (version 25).

## 3. Results

### 3.1. Spoken Languages

In the year 2018, N = 114 families were admitted to the pediatric palliative care unit. Of these, n = 49 (43.0%) did not speak the local language as their mother tongue. All following analyses and findings refer to this subgroup. For n = 44 (89.8%) parents an identification of spoken languages was possible, n = 4 parents spoke more than one language other than the local language (Table 1).

### 3.2. Local Language Skills

#### 3.2.1. Overall Distribution of Language Skills

For the medical staff, an assessment of language skills was possible for n = 38 (77.6%) mothers and n = 32 (65.3%) fathers. The psychosocial staff could assess n = 37 (75.5%) mothers and n = 26 (53.1%) fathers.

#### 3.2.2. Superordinate Competence Levels

According to the medical staff, 36.8% of mothers and 25.0% of fathers spoke at the A Level (lowest level), 21.1% of mothers and 21.9% of fathers at the B Level (medium level) and the majority of mothers (42.1%) and the majority of fathers (53.1%) at the C Level (highest level).

Overall, the psychosocial staff more frequently assigned poorer language levels to mothers and fathers. From their perspective, the majority of mothers (37.8%) and 34.6% of fathers spoke at the A Level (lowest level), 35.1% of mothers and 26.9% of fathers at the B Level (medium level) and 27.0% of mothers and the majority of fathers (38.5%) at the C Level (highest level). Figure 2ii,iv shows the distribution of the competence levels.

#### 3.2.3. Subordinate Language Skills Levels

Considering the two extremes of the language skills levels (A0 = no skills, C2 = mastery), the medical staff attributed the A0 Level to 7.9% of mothers and 6.3% of fathers. The C2 Level was attributed to 23.7% of mothers and 31.3% of fathers. For mothers, the three most frequently assigned language skill levels were C2 (23.7%), C1 (18.4%) and A0+ (18.4%). For fathers, levels C2 (31.3%), C1 (21.9%) and B1 (12.5%) were the most common.

According to the psychosocial staff, 18.9% of mothers and 15.4% of fathers spoke at the A0 Level; 16.2% of mothers and 19.2% of fathers spoke at the C2 Level. Mothers were most often assigned levels B2 (24.3%), A0 (18.9%) and C2 (16.2%). Levels C2 (19.2%), C1 (19.2%) and B2 (19.2%) were the three most common levels allocated to fathers. Figure 2i,iii shows the distribution of the language skills levels.

### 3.3. Interprofessional Agreement

For 37 (75.5%) individual mothers and 26 individual (53.1%) fathers, it was possible to determine an agreement between the assessments of the medical and the psychosocial staff.

#### 3.3.1. Superordinate Competence Levels

For 73.0% of the mothers it revealed an agreement between the two professions. Kappa statistics confirmed a moderate agreement between the two professions (*κ* = 0.59). When the professions coincided in their assessment, in 44.4% of the cases they jointly assigned an A-Level, in 22.2% a B-Level and in 33.3% a C-Level. Mothers, on whom the two professions disagreed, were mostly given the better competence levels by the medical staff (70%) in contrast to the psychosocial staff.

Among fathers, the two professions matched in 73.1% of individual cases (*κ* = 0.58, moderate convergence). Here, the professions consistently assigned an A Level in 36.8% of cases, a B Level in 15.8% and a C Level in 47.4% of cases.

In cases where the professions disagreed, it was again consistently the medical staff (71.4%) who assigned the better competence levels. Figure 3 shows the inter-professional agreement for mother’s and father’s competence levels.

#### 3.3.2. Subordinate Language Skills Levels

The two professions agreed in 40.5% of mothers regarding their language skills levels (Figure 3). The agreement between the two professions was fair (*κ* = 0.31). When assessments matched, the professions most frequently awarded the A0 Level (20.0%), the A0+ Level (20.0%) and the C2 Level (26.7%). When disagreeing, it was again mostly the medical staff who gave the better language skills levels to mothers (72.2%).

For 46.2% of fathers, the assessments of the two professions matched (*κ* = 0.36, fair; Figure 3). The most frequently coincidently assigned language levels were A0 (16.7%), A0+ (16.7%) and C2 (41.7%). In case of disagreement, the medical staff attributed better language skills levels to fathers than the psychosocial staff (78.6%).

## 4. Discussion

### 4.1. Main Findings

The results show that almost half of the parents on the pediatric palliative care unit speak a mother tongue other than the local language. This finding demonstrates on the basis of concrete data that language barriers seem to be pervasive in pediatric palliative care and is therefore generally consistent with other research [15,26,35,36]. At the same time, this implies the need for culturally sensitive communication in the health care sector and in the field of pediatric palliative care [22,30,37,38,39]. In our study, Turkish constituted the most frequently identified language. Considering that families of Turkish origin make up the largest group with a migration background in Germany, this finding seems to represent the population distribution typical for Germany [40]. However, language barriers faced by such a large national group also imply that critically ill children may not receive the best possible care. One essential care objective of pediatric palliative care, which is to take into account the whole family, can only be insufficiently fulfilled if, due to insufficient language skills, parents receive an inadequate picture of their child’s disease situation, are less able to participate in the treatment process of their child or are unable to express their own wishes or needs [7,13,15,39]. This last point can also complicate good interpersonal relationship building and the important joint collaboration between healthcare professionals and parents.

According to both professions, more fathers speak at the highest language levels than mothers. This may be because mothers commonly care for the sick child, while fathers have a job and may therefore experience more language-promoting contacts [41,42].

Interestingly, the assessments of the two professions concerning parental language skills were only partly consistent. Overall, the medical staff ascribed almost mother-tongue skills to both the majority of mothers and fathers, whereas the psychosocial staff tended to ascribe worse local language skills to the majority.

On the individual level, in a direct comparison of the professions, it was consistently the medical staff that attributed better language skills to mothers and fathers.

Regarding medical and nursing issues, many situations require clear information and instructions, e.g., “Give this medicine to your child every morning”. If a parent follows such instructions, the physician or nurse may assume that they understood because no detailed conversation is necessary. Further, it is possible that over time, parents acquire the “technical medical language”, but in contexts that are not relevant to their child’s care, they may have a significantly lower level of local language skills. Nevertheless, the exchange with parents in the care context is the medical staff’s reference for the assessment and could therefore explain why they ascribed good language skills to the parents overall.

It is plausible that parents’ language skills were rated worse by psychosocial staff because the contents of the joint contacts with parents are completely different. They include verbal exchanges of feelings, emotions and needs. For non-native speakers, this is a highly complex requirement. First, the concretization of emotion is culture-specific, which can trigger misunderstandings if emotions are expressed differently in the local language [43,44]. Second, thoughts and worries are often difficult to describe in one’s own language. Expressing these in a foreign language is even more challenging and reveals existing language deficits. Third, mental illness is stigmatized in many cultures [45,46,47]. This can result in a parent refusing to talk to a psychologist, and thereby, giving the impression that he or she cannot follow the conversation. In summary, it is therefore conceivable, that non-native-speaking parents have problems expressing their emotions in a foreign language, are not understood by professionals, do not dare to speak about emotions or are worried about being stigmatized.

Besides the disagreement in judging the parents’ language skills, it is worth noting that the professions often coincided in their assessment of an individual parent. The agreement between the professions was consistently based either on the common attribution of rather poor or rather good language skills. This seems to suggest that professions within an institution perceive parents in a similar way especially when parent’s local language skills are neither conspicuously good nor bad.

### 4.2. Implications for Clinical Practice

Many parents in pediatric palliative care are not proficient in the local language, leading to language barriers. This is especially pronounced in mothers, who are the main caregiver of the sick child. Therefore, an institution should develop strategies to deal with this situation. One solution may be to regularly include fathers in conversations with the care team. They could pass on the contents of the discussions to their wives and thus could serve as a valuable resource in understanding the cultural peculiarities of a family.

Another approach may be to involve interpreters not only for fixed appointments but also in everyday life on a unit. This way, a brief exchange between professionals and parents could be facilitated. However, the communication about emotions and the establishment of a good relationship with parents remains complex when interpreters must be integrated into conversations [19].

Our results show that staff members from different professions do not necessarily agree in their assessment of parental local language skills. Professionals should be aware of this fact and exchange their impressions and assessments in multi-professional team meetings to avoid inconsistencies or even mistakes in daily care. At the same time, however, this shows the importance of multi-professional pediatric palliative care: the perspective of just one profession on patients’ caregivers is inadequate and is obviously biased by subjective and profession-dependent factors. Regular multi-professional team meetings should therefore be initiated, particularly on families with a different ethnic background, in which the assumptions of individual persons involved in the care process are discussed. For the communication between parents and the professional care team, professional translators should participate by default in all discussions, or even accompany the parents throughout their stay. The recruitment of multilingual team members from different cultures can also be a way to overcome language and cultural barriers [20].

Besides the language barriers, cultural differences need to be considered. The success of intercultural communication does not solely depend on the mastery of a common language, but is significantly determined by the knowledge of cultural aspects [22]. Cultural factors can, for example, determine what parents want or don’t want to talk about with professionals, which treatment methods are accepted, the preferred gender of the discussion partner, or the position a particular profession holds for parents [13,48,49]. A culture-specific understanding of disease might therefore impact the conversation with the care team. In Islamic belief, for example, it is assumed that diseases are caused in Allah’s knowledge and permission and are often understood as a test [50]. Understanding these cultural features is essential in properly understanding the families. Therefore, staff should be sensitized for it.

### 4.3. Limitations of the Study

This study is a first step towards an in-depth knowledge of the linguistic and cultural characteristics in pediatric palliative care. Nevertheless, limitations of the study must be pointed out. The study was conducted solely in Germany over a recruitment period of one year. The results should, therefore, not be regarded as representative of the entire clientele of the institution, nor of the numerous different settings of pediatric palliative care. Further studies in different countries and settings, over a longer time period, are necessary to be able to make more general statements about language characteristics of parents. Nevertheless, this study should be seen as a prelude to research on an enormously important but underrepresented topic in pediatric palliative care.

The greatest limitation of the study is that the spoken language of parents alone does not allow conclusions to be drawn about cultural differences, behaviors or views. The interpretations presented above are therefore only to be understood as explanatory approaches and should be explored in future studies.

## 5. Conclusions

The present study shows that pediatric palliative care institutions must attune to many families who do not speak the local language. This finding contains important information for treatment and study planning in this highly complex setting. Language barriers are always an indication of cultural barriers, which must be identified and brought to the attention of the numerous multi-professional providers of pediatric palliative care. If numbers on spoken languages in an institution are known, documents can, for example, be translated into certain languages, a pre-selection of interpreters can be made, or culturally-related training courses can be planned. Overall, language-related figures provide an indication of the cultures to which an institution must adapt. Suitable measures can thus be selected and initiated more efficiently. Successful communication is essential to ensure the best possible care for the seriously ill children treated in pediatric palliative care, to avoid treatment errors and to meet the needs of the families. This also means that the different professions are aware that their own assessment and perception of parents does not have to agree with other colleagues which can lead to significant misunderstandings. Overall, our study shows that an abstract awareness of language barriers in pediatric palliative care is not sufficient. Language barriers can only be efficiently countered by concrete data and the findings derived from them. Overall, we therefore propose the following agenda, to implement culturally sensitive pediatric palliative care in the long term: (a) collecting national and international data on languages spoken in pediatric palliative care, (b) based on these data and complementary research projects, relevant cultures in pediatric palliative care must be identified, (c) relevant characteristics and the dealing with the cultures identified should be integrated into education and training curricula for pediatric palliative care, and (d) in this context, pediatric palliative care providers need to be made aware of the different perception bias of individuals and professions.

## Figures and Tables

**Figure 1 children-07-00118-f001:**
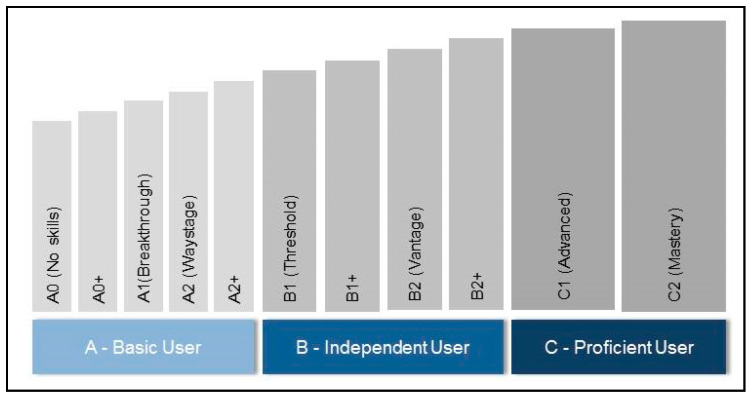
Adapted version of the “Common European Framework of Reference for Languages” as used for study purpose.

**Figure 2 children-07-00118-f002:**
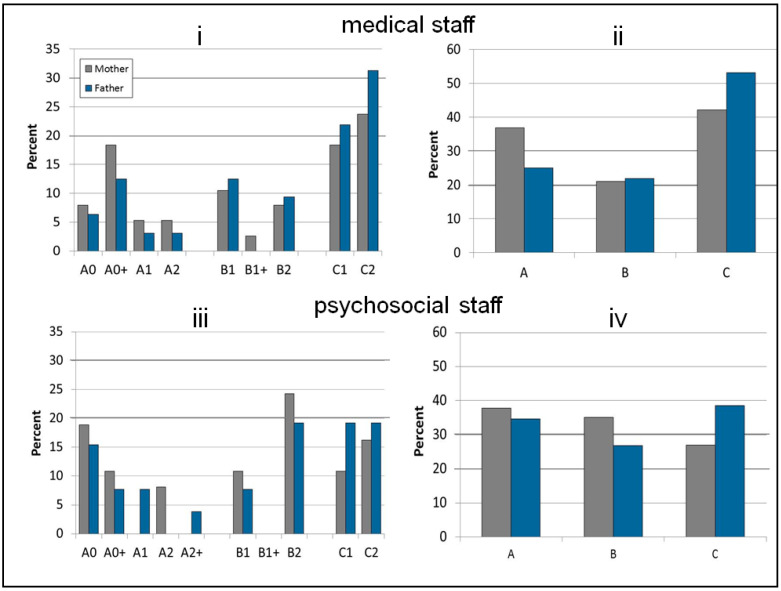
Overall distributions of language skills levels (**i**,**iii**) and competence levels (**ii**,**iv**) of mothers and fathers as assessed by medical and psychosocial staff.

**Figure 3 children-07-00118-f003:**
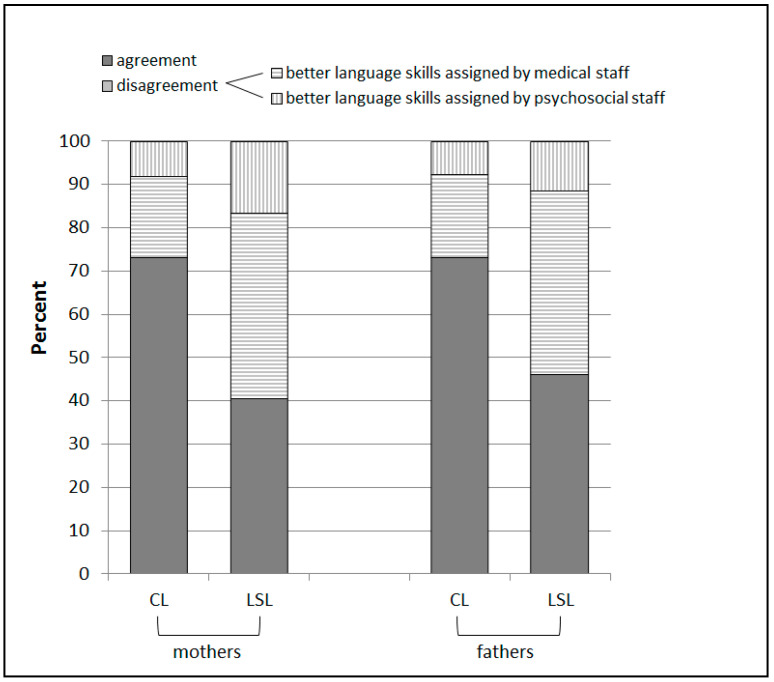
Inter-professional agreement on mother’s and father’s competence levels (CL) and language skills levels (LSL) and information on which profession assigned better levels in the event of disagreement (striped).

**Table 1 children-07-00118-t001:** Spoken languages of parents as indicated by the participants (multiple entries possible).

Language	n	%
Turkish	22	45.8
Arabic	6	12.5
Albanian	3	6.3
Syriac	2	4.2
Italian	2	4.2
Croatian	2	4.2
Russian	2	4.2
Kurdish	2	4.2
Lebanese	1	2.1
Kazakh	1	2.1
Persian	1	2.1
Romanian	1	2.1
Sinti	1	2.1
Armenian	1	2.1
**Total**	**48**	**100**

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
