# Peer review of "Quantifying the Language Barrier—A Total Survey of Parents’ Spoken Languages and Local Language Skills as Perceived by Different Professions in Pediatric Palliative Care"

_children, 2020, doi:10.3390/children7090118_

Round 1

Reviewer 1 Report

Overall, this is a good paper on an important topic. It could be strengthened by adding more discussion on the implications of this topic and strategies for improving coordinated care, care team communication, and care team-family communication. An agenda for next steps would also be a useful addition. 

Author Response

Thank you very much for your comments and the opportunity to revise our paper with regard to your suggestions. We have added the mentioned aspects and an agenda to the discussion:

  • 6, ll. 238-240; “This finding demonstrates on the basis of concrete data that language barriers seem to be pervasive in pediatric palliative care and is therefore generally consistent with other research [15,26,35,36].”
  • 6/7, ll. 244-251; “However, language barriers faced by such a large national group also imply that critically ill children may not receive the best possible care. One essential care objective of pediatric palliative care, which is to take into account the whole family, can only be insufficiently fulfilled if, due to insufficient language skills, parents receive an inadequate picture of their child's disease situation, are less able to participate in the treatment process of their child or are unable to express their own wishes or needs [7,13,15,39]. This last point can also complicate good interpersonal relationship building and the important joint collaboration between healthcare professionals and parents.”
  • 8, ll. 301-309; “At the same time, however, this shows the importance of multi-professional pediatric palliative care: the perspective of just one profession on patients’ caregivers is inadequate and is obviously biased by subjective and profession-dependent factors. Regular multi-professional team meetings should therefore be initiated, particularly on families with a different ethnic background, in which the assumptions of individual persons involved in the care process are discussed. For the communication between parents and the professional care team, professional translators should participate by default in all discussions, or even accompany the parents throughout their stay. The recruitment of multilingual team members from different cultures can also be a way to overcome language and cultural barriers [20]”.
  • 8, ll. 310-315; “The success of intercultural communication does not solely depend on the mastery of a common language, but is significantly determined by the knowledge of cultural aspects [22]. Cultural factors can, for example, determine what parents want or don't want to talk about with professionals, which treatment methods are accepted, the preferred gender of the discussion partner, or the position a particular profession holds for parents [13,48,49].”
  • 8, ll. 336-338; “Language barriers are always an indication of cultural barriers, which must be identified and brought to the attention of the numerous multi-professional providers of pediatric palliative care.”
  • 8, ll. 340-342; “Overall, language-related figures provide an indication of the cultures to which an institution must adapt. Suitable measures can thus be selected and initiated more efficiently.”
  • 8/9, ll. 346-355; “Overall, our study shows that an abstract awareness of language barriers in pediatric palliative care is not sufficient. Language barriers can only be efficiently countered by concrete data and the findings derived from them. Overall, we therefore propose the following agenda to implement culturally sensitive pediatric palliative care in the long term: (a) collecting national and international data on languages spoken in pediatric palliative care, (b) based on these data and complementary research projects, relevant cultures in pediatric palliative care must be identified, (c) relevant characteristics and the dealing with the cultures identified should be integrated into education and training curricula for pediatric palliative care, (d) In this context, pediatric palliative care providers need to be made aware of the different perception bias of individuals and professions.”

Reviewer 2 Report

Dear Authors.

I think you have raised an interesting question about communication and culture. However I miss some explanation abot thrir relation in the introduction. I would like you to develop on how they are related to eatchother and how they are used in an everyday clinical ward today. 

I also would like you to clarifye to the redaer hwo the medical staff was and who the nursing staff or other staff were along with a dscription of their languagae skills. 

I would like you to describe the clinical setting more thorough as well as if a translator are used at some point or not.

In the discussion section I would like you to abstract your findings a bit more and relate it to the things you write about in the introduction. This to frame the findings and give them a context. I also would like you to elaborate on the importance of your findings.

thank you for this important study 

Author Response

Reviewer 2: Dear Authors.

I think you have raised an interesting question about communication and culture. However I miss some explanation about their relation in the introduction. a) I would like you to develop on how they are related to each other and how they are used in an everyday clinical ward today.

I also would like you to b) clarify to the reader who the medical staff was and who the nursing staff or other staff were along with a description of their language skills.

I would like you to c) describe the clinical setting more thorough as well as if a translator are used at some point or not.

In the discussion section I would like you to d) abstract your findings a bit more and relate it to the things you write about in the introduction. This to frame the findings and give them a context. I also would like you to elaborate on the importance of your findings.

thank you for this important study

Thank you very much for your comments and the opportunity to revise our paper with regard to your suggestions. Since you are addressing various aspects that require revision, we have included letters in your text inserted above to make changes more transparent and refer to these in the following.

a) Thank you for this comment. We added the required information on communication and culture at various points in the introduction:

  • p. 1, l. 34-37; “Language and culture form an inseparable, complex and mutually influential unit. Language is formed by the unique characteristics of a culture and at the same time gives expression to it. Comparable to a mirror, language therefore provides essential insights into the culture of a person [3,4].”
  • p. 2, ll. 53-60; “In the clinical routine in general and particularly in pediatric palliative care, different strategies such as the involvement of (professional) translators are used to overcome language barriers between care providers and patients/families [13,18-25]. The importance of the awareness and education of cultural characteristics is acknowledged by many healthcare providers, but is perceived as insufficient [26-28]. However, successful intercultural communication can only be achieved if providers develop culturally sensitive skills and hence act on the same linguistic and superordinate cultural level as parents and patients with a different ethnic background [22,29-31].”
  • p. 2, ll. 62-64; “However, a thorough knowledge of the languages and language skills of parents can provide a first significant insight into the cultures encountered in pediatric palliative care and thus contribute to culturally sensitive care.”

b) We have added the requested details on the staff groups (p.3, ll. 109-111; […] 1) a member of the medical staff (physicians, nurses) and 2) a member of the psychosocial staff (psychologist, pedagogue) was obtained for each parent. All staff members were native speakers of German (the local language).”

c) Thank you for this comment. We have integrated the information on the clinical setting into the paragraph on information on the study participants (p. 2, ll. 71-76; “Clinical setting and participants; The study was conducted on a pediatric palliative care unit which is affiliated to a pediatric hospital in a national conurbation of Germany. A multi-professional team annually cares for about N=170 children and adolescents between 0-21 years with life-limiting diseases in a total of 8 patient rooms [32]. Additional services such as the occasional use of interpreters, especially for conversations between parents and professionals, complete the care spectrum.”

d) Thank you for this comment. We have added the mentioned aspects to the discussion:

  • p. 6, ll. 238-240; “This finding demonstrates on the basis of concrete data that language barriers seem to be pervasive in pediatric palliative care and is therefore generally consistent with other research [15,26,35,36].”
  • pp. 6/7, ll. 244-251; “However, language barriers faced by such a large national group also imply that critically ill children may not receive the best possible care. One essential care objective of pediatric palliative care, which is to take into account the whole family, can only be insufficiently fulfilled if, due to insufficient language skills, parents receive an inadequate picture of their child's disease situation, are less able to participate in the treatment process of their child or are unable to express their own wishes or needs [7,13,15,39]. This last point can also complicate good interpersonal relationship building and the important joint collaboration between healthcare professionals and parents.”
  • p. 8, ll. 301-309; “At the same time, however, this shows the importance of multi-professional pediatric palliative care: the perspective of just one profession on patients’ caregivers is inadequate and is obviously biased by subjective and profession-dependent factors. Regular multi-professional team meetings should therefore be initiated, particularly on families with a different ethnic background, in which the assumptions of individual persons involved in the care process are discussed. For the communication between parents and the professional care team, professional translators should participate by default in all discussions, or even accompany the parents throughout their stay. The recruitment of multilingual team members from different cultures can also be a way to overcome language and cultural barriers [20]”.
  • p. 8, ll. 310-315; “The success of intercultural communication does not solely depend on the mastery of a common language, but is significantly determined by the knowledge of cultural aspects [22]. Cultural factors can, for example, determine what parents want or don't want to talk about with professionals, which treatment methods are accepted, the preferred gender of the discussion partner, or the position a particular profession holds for parents [13,48,49].”
  • p. 8, ll. 336-338; “Language barriers are always an indication of cultural barriers, which must be identified and brought to the attention of the numerous multi-professional providers of pediatric palliative care.”
  • p. 8, ll. 340-342; “Overall, language-related figures provide an indication of the cultures to which an institution must adapt. Suitable measures can thus be selected and initiated more efficiently.”
  • p. 8/9, ll. 346-355; “Overall, our study shows that an abstract awareness of language barriers in pediatric palliative care is not sufficient. Language barriers can only be efficiently countered by concrete data and the findings derived from them. Overall, we therefore propose the following agenda to implement culturally sensitive pediatric palliative care in the long term: (a) collecting national and international data on languages spoken in pediatric palliative care, (b) based on these data and complementary research projects, relevant cultures in pediatric palliative care must be identified, (c) relevant characteristics and the dealing with the cultures identified should be integrated into education and training curricula for pediatric palliative care, (d) In this context, pediatric palliative care providers need to be made aware of the different perception bias of individuals and professions.”

Round 2

Reviewer 2 Report

Thanks for your interesting paper that I now find sound and clear